# PD-L1 Expression in Patients with Idiopathic Pulmonary Fibrosis

**DOI:** 10.3390/jcm10235562

**Published:** 2021-11-26

**Authors:** Sissel Kronborg-White, Line Bille Madsen, Elisabeth Bendstrup, Venerino Poletti

**Affiliations:** 1Center for Rare Lung Diseases, Department of Respiratory Diseases and Allergy, Aarhus University Hospital, 8200 Aarhus, Denmark; karbends@rm.dk (E.B.); venerino.poletti@gmail.com (V.P.); 2Department of Pathology, Aarhus University Hospital, 8200 Aarhus, Denmark; linemads@rm.dk; 3Department of the Diseases of the Thorax, Ospedale Morgagni, University of Bologna, 47121 Forli, Italy

**Keywords:** idiopathic pulmonary fibrosis, cryobiopsy, PD-L1

## Abstract

Background: Idiopathic pulmonary fibrosis (IPF) is the most common and severe form within the group of idiopathic interstitial pneumonias. It is characterized by repetitive alveolar injury in genetically susceptible individuals and abnormal wound healing, leading to dysregulated bronchiolar proliferation and excessive deposition of extracellular matrix, causing complete architectural distortion and fibrosis. Epithelial-to-mesenchymal transition is considered an important pathogenic event, a phenomenon also observed in various malignant neoplasms, in which tumor cells express programmed death-ligand one (PD-L1). The aim of this study was to assess the presence of PD-L1 in patients with IPF and other interstitial lung diseases (ILDs). Method: Patients with a clinically and radiologically suspected idiopathic interstitial pneumonia or other ILDs undergoing transbronchial cryobiopsy to confirm the diagnosis at the Department of Respiratory Diseases and Allergy, Aarhus University Hospital, were included in this prospective observational study. Cellular membrane PD-L1 expression in epithelial cells was determined using the DAKO PD-L1 IHC 22C3 PharmDx Kit. Results: Membrane-bound PD-L1 (mPD-L1) was found in twelve (28%) of the forty-three patients with IPF and in five (9%) of the fifty-five patients with other ILDs (*p* = 0.015). When adjusting for age, gender and smoking status, the odds ratio of having IPF when expressing mPD-L1 in alveolar and/or bronchiolar epithelial cells was 4.3 (CI: 1.3–14.3). Conclusion: Expression of mPD-L1 in epithelial cells in the lung parenchymal zones was detected in a consistent subgroup of patients with IPF compared to other interstitial pneumonias. Larger studies are needed to explore the role of mPD-L1 in patients with IPF.

## 1. Introduction

Idiopathic pulmonary fibrosis (IPF) is the most common and severe form of idiopathic interstitial pneumonia, with a mean survival time of 3–5 years without antifibrotic treatment. Patients suffering from IPF have an increased risk of peripheral lung cancer [1], with a reported prevalence between 7 and 31% [1,2]. The phenotypes and genetic alterations of peripheral lung cancer associated with IPF suggest that IPF and lung cancer may share several pathogenic pathways [3,4,5].

IPF is characterized by a progressive scarring of the lung parenchyma due to dysregulated bronchiolar proliferation and excessive accumulation of extracellular matrix (ECM), resulting in architectural distortion and respiratory failure. The pathogenesis of these changes is still not fully understood, but it may be divided into three steps. The first step is microinjuries occurring in type 1 pneumocytes from environmental exposures (e.g., smoking, air pollution, occupational exposures) [6,7,8] and mechanical stress [9]. The second step is linked to the presence of alveolar stem cell exhaustion. In predisposed subjects, the type II alveolar epithelial cells are unable to respond appropriately to the recurrent alveolar microinjury due to intrinsic factors (e.g., mutations of genes controlling the telomere length, the production of surfactant proteins) or of extrinsic factors (smoking, pollution) [10,11,12]. These senescent cells acquire the so-called senescent-associated secretory phenotype (SASP) and thereafter activate a variety of immune and inflammatory pathways (i.e., Wnt/β-catenin pathway, TGF-β, Sonic Hedgehog, Notch) [13,14,15,16]. In the alveolar areas, SASP type II pneumocytes induce a profibrotic microenvironment with the appearance of fibroblastic foci. The third step is when this wave of injuries reaches the centriacinar areas. Bronchiolar stem cells that have an intrinsic anti-apoptotic machinery, such as the expression of truncated protein p63, start to proliferate and migrate. The result is a dysplastic bronchiolar proliferation (honeycombing changes), myofibroblast proliferation and collagen deposition, leading to a complete lung parenchymal derangement.

In recent years, the discovery of immune checkpoint inhibitors for the treatment of patients with various types of cancer, including lung cancer, has revolutionized the prognosis of patients with cancer [17]. One of the immune checkpoint receptors, programmed death-1 (PD-1), is mainly expressed in activated T and B cells and is an inhibitory molecule controlling inflammatory responses to injury. PD-1 binds to two different ligands, programmed death-ligand 1 (PD-L1) and programmed death-ligand 2 (PD-L2), which can be found in a wide variety of cells, including different immune cells and epithelial cells [18,19]. The PD-1/PD-L1 axis plays an important role in regulating and limiting the effector T cells’ response in peripheral tissues, as a negative feedback loop, in relation to infection and inflammation. Moreover, this axis also plays an important role in diminishing autoimmunity. In some types of cancer, the tumor cells express PD-L1, and to a lesser extent PD-L2, as a way to evade detection or even inhibit the immune response [20,21], which is crucial for the unrestricted growth of tumor cells. Once activated, the PD-L-positive T cells block further activation and proliferation. Immune checkpoint inhibitors, such as anti-PD-1 and anti-PD-L1, work by preventing the immune checkpoint molecule from being activated, enabling T cells to proliferate and thereby attack the tumor cells. Upregulation of PD-L1 expression in the membrane of the epithelial cells in a variety of malignant tumors has been shown to promote epithelial-to-mesenchymal transition (EMT). In the EMT process, polarized epithelial cells undergo changes to become mesenchymal stem cells, and they gain functions such as migration and invasion, elevated resistance to apoptosis and increased production of ECM components [22,23]. EMT is also considered a key feature in the pathogenesis of IPF [16,24].

In a recent pilot study, Jovanovic et al. showed that soluble PD-L1 serum levels in patients with IPF were significantly higher compared to healthy controls [25]. Moreover, they showed an overexpression of membrane PD-L1 on alveolar macrophages in surgical lung biopsies in nine of twelve patients with IPF. Soluble PD-L1 could be a prognostic or/and predictive biomarker of the disease [26]. However, data on the expression of this marker on epithelial cells that are detectable in the fibrotic areas in lung tissue obtained from patients with IPF are still scarce.

Similarly, Celada et al. found that PD-1 expression on the surface of circulating CD4+ T cells was significantly higher in patients with IPF compared to healthy controls. Immunohistochemical staining of PD-L1 and PD-1 in lung biopsies from patients with IPF and healthy controls showed increased expression in lymphocytes in 12 patients [26]. In another study, PD-1, but not PD-L1, was increased on T cells in both peripheral blood and lung biopsies from patients with IPF compared to healthy controls [27]. 

The aim of the present study was to analyze the expression of PD-L1 in epithelial alveolar and/or bronchiolar cells present in fibrotic areas in lung tissue obtained through cryobiopsies from patients with IPF and compare these to patients with other interstitial lung diseases (ILD). Analysis of blood concentration of soluble PD-L1 was not considered for this study, but blood samples were collected and stored for further investigation. 

## 2. Methods

### 2.1. Patients

Patients having cryobiopsies performed as part of an investigation for ILD at the Center for Rare Lung Diseases, Department of Respiratory Diseases and Allergy, Aarhus University Hospital, Denmark, were enrolled in the study. Only patients >18 years providing written informed consent could participate. Biopsies were not taken in patients with a documented significant lung function impairment (forced vital capacity (FVC) < 50%, lung transfer factor for carbon monoxide (TLco) < 35%), if body mass index was >35, if pulmonary hypertension with tricuspid gradient above 40 mmHg was evident or if the patient was suffering from other cardiac diseases or had other comorbidities increasing the risk of complications. Patients were also excluded if the cryobiopsy procedure was not possible for technical reasons reaching the high-resolution computed tomography (HRCT) scan selected area. 

### 2.2. Data Collection

A HRCT pulmonary function test (PFT) (FVC, TLco), analyzed in accordance with the Global Lung Function Initiative reference values [28], and the six-minute walk test (6MWT) were performed at referral or at the last visit before the procedure. Baseline demographics on gender, age and smoking history were collected. At the one-year follow-up, PFT and 6MWT were performed and absolute changes were used for analysis. 

### 2.3. Cryobiopsy Procedure

The cryobiopsy procedure was performed as previously described [29]. Patients were under general anesthesia using propophol, remifenthanyl and rocuronium and intubated with a Rusch orotracheal tube. Bronchoalveolar lavage (BAL) was performed in all patients before the procedure. Four cryobiopsy samples from two different segments were obtained when possible, using a 1.9 or 2.4 mm cryoprobe (ERBE, Tübingen, Germany) under fluoroscopic guidance. The more affected areas were selected based on the basis of HRCT scan features.

The frozen lung samples were thawed in isotonic saline, fixed in formalin and finally embedded in paraffin in accordance with standard procedures. 

### 2.4. Pathology Method

All specimens were serially sectioned. The first section was stained with hematoxylin–eosin; the others were used for immunohistochemistry. The pathologic report followed previously published criteria [30]. A report for each sample was obtained including site, size of the specimen, characteristics for central versus peripheral sampling and the histopathologic pattern identified. The final diagnostic report was made including data for each sample. Evaluation of the PD-L1 protein expression was determined by use of the DAKO PD-L1 IHC 22C3 pharmDx kit (Agilent, Santa Clara, CA, USA). The slides were screened at low power, and the expression of membrane-bound PD-L1 (mPD-L1) in the epithelial cells present in fibrotic lung zones (excluding epithelial cells covering histologically normal bronchioles) was finally confirmed at high power (×40). The amount of positive mPD-L1 (mPD-L1_pos_) parenchymal epithelial cells (type II pneumocytes or bronchiolar cells) was graded using a semiquantitative approach: <25%, 25–49%, 50–74% and 75–100% of the epithelial alveolar and/ or bronchiolar cells. The intensity was graded in three categories (1 = weak, 2 = moderate, 3 = strong). The pathologist was blinded to the final diagnosis.

### 2.5. Statistics

Normally distributed data are presented as mean values ± standard deviation (SD), while data with a non-normal distribution are presented as median values with interquartile ranges (IQR) or total range. Data were checked for normality by histogram and the quantile–quantile plot. Continuous data were analyzed with the unpaired *t*-test or the Mann–Whitney test, as appropriate, according to their distribution. Categorical data are presented as proportions of the total population and analyzed with the chi^2^-test or Fisher’s exact test when the total number of one group was ≤5. A *p*-value of <0.05 was considered statistically significant. Stata IC (version 16.0, StataCorp, College Station, TX, USA) was used for statistical analysis.

## 3. Results

### 3.1. Demographics

Between September 2017 and May 2019, 102 patients were enrolled in the study. Four patients were excluded (three patients because biopsies were not obtained; one patient withdrew consent). IPF was diagnosed in 43 (44%) patients, and 55 (56%) patients were diagnosed with other ILDs than IPF (non-IPF), e.g., twenty with hypersensitivity pneumonitis (HP), seven with non-specific interstitial pneumonitis (NSIP), two with desquamative interstitial pneumonitis (DIP), one with eosinophilic pneumonia and twenty-five with other ILDs. Baseline characteristics are presented in Table 1. Higher age (*p* = 0.047) and traction bronchiectasis (*p* < 0.001) were identified more often in patients with IPF. Traction bronchiectasis was also found more frequently in mPD-L1_pos_ IPF patients compared to the mPD-L1_pos_ non-IPF patients (Table 2) (*p* = 0.003). When grouping the patients into mPD-L1_pos_ and mPD-L1-negative (mPD-L1_neg_) (Table 3), baseline characteristics were not significantly different. TLco was significantly lower (*p* = 0.007) and GAP index 2 more frequent in the mPD-L1_pos_ IPF group compared to the mPD-L1_neg_ IPF group (see Table 4). When adjusting for age, gender and smoking status, the odds ratio for IPF when expressing mPD-L1 was 4.3 (CI: 1.3–14.3). The median number of days between first visit and the cryobiopsy procedure was 55 days (IQR: 27–82).

### 3.2. Specific Histological Lesions

The specific histological lesions identified are reported in Table 5. In the IPF group, 12 cases (28%) were mPD-L1_pos_ in the epithelial alveolar and/or bronchiolar cells. In the non-IPF group, five cases were found (9%) (*p* = 0.015). Two of the non-IPF patients were diagnosed with HP, two with NSIP and one with DIP. The intensity of the mPD-L1 staining was weak (grade 1) in four of the patients and moderate (grade 2) in thirteen of the patients. In one patient, mPD-L1-positive cells were found in <25%, in six patients in 25–49%, in eight patients in 50–74% and in two patients in >75% of the cells (Figure 1). As expected, macrophages also expressed PD-L1. 

When comparing the mPD-L1_pos_ and mPD-L1_neg_ patients, we found that fibroblast foci (*p* = 0.028) and a combination of patchy fibrosis and fibroblast foci *p* = 0.02) were more frequently detected in the mPD-L1_pos_ patients. When comparing the IPF mPD-L1_pos_ patients with the IPF mPD-L1_neg_ patients, or the IPF mPD-L1_pos_ with the non-IPF mPD-L1_pos_ patients, we found no differences in histological elementary lesions. We found no association between honeycombing and expression of mPD-L1. No association was found either between lymphocytosis in BAL (lymphocytes > 15%) and PD-L1_pos_ patients (*p* = 0.97) or between percentages of macrophages in BAL and PD-L1_pos_ patients (*p* = 0.36), nor did we find an association between C-reactive protein levels and PD-L1_pos_ patients (*p* = 0.66).

### 3.3. One-Year Follow-Up

During the one-year follow-up, four patients died (two IPF mPD-L1_neg_, one IPF mPD-L1_pos_ and one non-IPF mPD-L1_neg_). Four patients withdrew from further participation in the study (one with IPF and three non-IPF). There was a small but significant difference in the change in TLco at the one-year follow-up between the patients with IPF (−1.5 ± 8) and the non-IPF patients (5.2 ± 12) (*p* = 0.003). There were no other significant differences in PFT or distance in 6MWT when comparing patients with IPF to non-IPF patients, mPD-L1_pos_ patients to mPD-L1_neg_ patients, mPD-L1_pos_ IPF patients to mPD-L1_neg_ IPF patients or PD-L1_pos_ IPF to PD-L1_pos_ non IPF patients (Table 6). Of the patients diagnosed with IPF, 40 (93%) started antifibrotic treatment (24 patients pirfenidone, 16 patients nintedanib). Four of the five mPD-L1_pos_ non-IPF patients were treated with immunosuppressants (steroids and azathioprine). 

## 4. Discussion

The present study describes the expression of mPD-L1 in alveolar and bronchiolar epithelial cells in a subgroup of patients with IPF. The expression of PD-L1 and PD-1 in cells of the immune system in patients with IPF has so far only been reported in a few studies [25,26]. Although we also found mPD-L1 in a few patients with other ILDs, mPD-L1 was found significantly more often in patients with IPF. Interestingly, IPF patients who were mPD-L1_pos_ had lower TLco, and more were never-smokers compared to the mPD-L1_neg_ IPF patients. TLco seemed to be lower, though not statistically significant, in the combined group of IPF and non-IPF mPD-L1_pos_ patients compared to the mPDL1_neg_ patients. Based on our data, it could be speculated that the expression of mPD-L1 may be associated with a peculiar subset of both IPF and non-IPF ILD patients, mainly a subset of IPF patients in whom fibroblast foci, or a combination of patchy fibrosis and fibroblast foci, are more densely represented compared to the mPD-L1_neg_ patients. There was no association between honeycombing changes and expression of mPD-L1. This could be explained by the size of the biopsies or the site where the biopsies were taken, taking into account that areas with honeycombing changes visualized on HRCT were deliberately avoided. Alternatively, it could be explained by the fact that epithelial cells expressing mPDL-1 are in a context of epithelial mesenchymal transition, as suggested by Yamaguchi M. et al. [31], therefore in zones where clear honeycomb changes have not fully developed.

Traction bronchiectasis, as a radiologic marker of fibrotic changes, was found more frequently in patients with IPF compared to non-IPF patients and also more often in mPD-L1_pos_ IPF compared to mPD-L1_pos_ non-IPF patients (Table 1 and Table 2). However, no difference was found when comparing mPD-L1_pos_ to mPD-L1_neg_ patients or within the group of IPF patients. We did not investigate the extent or severity of the fibrotic changes on HRCT to differentiate between mild, moderate and severe disease, which might have revealed differences between the groups.

During the one-year follow-up of the participants, there was a significant difference, although small, in the change in TLco in patients with IPF compared to non-IPF patients. Based on our results, we were unable to demonstrate that the expression of mPD-L1 was associated with a worse prognosis in patients with IPF or non-IPF ILD. This might be explained by the short duration of the follow-up period of only one year and because the majority of the patients with IPF (93%) were treated with antifibrotics. At present, we do not know if the presence of mPD-L1 is a key step in a subset of patients with IPF or a marker associated with disease development or progression. Moreover, we do not know if the expression has a protective role in preventing further progression and fibrosis or if it leads to more fibrotic changes.

In recent years, the discussion of lumping and splitting patients with fibrotic ILDs and IPF has been frequently debated. Several studies have suggested that these diseases share common traits with respect to clinical course and treatment response to antifibrotics, probably due to different triggers resulting in the initiation of common fibrotic pathways [32,33,34,35]. When looking at fibrotic diseases from this perspective, the occurrence of mPD-L1 could be associated with fibrotic changes and not IPF as a disease entity. The lack of difference in baseline characteristics and changes in PFT and 6MWT at the one-year follow-up of patients with IPF and non-IPF mPD-L1_pos_ patients might support this hypothesis, although traction bronchiectasis was found more frequently in the IPF mPD-L1_pos_ patients. However, the number of patients in each group was low, and thus we cannot draw any conclusions. On the other side, the uncovering of pathological processes leading to disease development and progression might identify ILD subtypes based on different pathways. Splitting the diseases opens up the possibilities of targeted treatment and precision medicine, as has been seen in cancers. 

Until now, IPF has been regarded as a risk factor for lung toxicity with respiratory deterioration and acute exacerbations when considering anti-PD-1 or anti-PD-L1 immunotherapy in non-small cell lung cancer. The incidence of any grade of pneumonitis was found to be 1.3–3.6% in a meta-analysis by Khunger et al. [36]; in a recent real-life study, the incidence was as high as 9.5% [37]. However, immunotherapy has reportedly been given to few patients with IPF and lung cancer without worsening their respiratory condition or showing any signs of lung toxicity [38,39,40]. Thus, it could be hypothesized that treatment with anti-PD-L1 or anti-PD-1 immunotherapy could be a promising future treatment in patients with IPF and mPD-L1 expression. One of the most feared adverse events of immunotherapy in IPF patients would be drug-induced pneumonitis. Adverse effects if treated with immunotherapy have to be anticipated and need to be handled similarly to patients with cancer [41]. 

Identification of a subgroup of patients in which expression of PD-L1 on membranes of epithelial cells in the injured lung areas could incite clinical trials with drugs already used to treat a variety of malignant tumors.

There are limited data regarding PD-L1 and PD-1 expression in non-cancer patients. However, a recent study found an increased expression of PD-L1 in renal cells in a mouse after certain types of injuries [42].

Our study has several limitations. First, the length of the follow-up time might have been too short to observe major disease progression. Most of the patients with IPF were diagnosed with well-preserved FVC and were treated with antifibrotics, which affects disease progression. Second, we were not able to perform a power calculation, and this may have affected the subgroup analysis in mPD-L1_pos_ IPF and non- IPF patients.

Our study also has several strengths. Cryobiopsies were performed in all patients by three experienced bronchoscopists, ensuring a high biopsy quality, and all patients were diagnosed after MDT conferences. Additionally, the pathologist (L.B.M.) was blinded to the final diagnosis, routinely assesses PD-L1 expression in lung cancer and is familiar with this analysis using a semiquantitative approach. All subjects, apart from one patient, agreed to participate in the study, which minimized selection bias. 

## 5. Conclusions

We demonstrated that the expression of mPD-L1 in epithelial cells in the alveoli and bronchiolar cells present in fibrotic areas was evident in a subgroup of patients with IPF. Larger studies with longer follow-up time are needed to explore the role of mPD-L1 in patients with IPF.

## Figures and Tables

**Figure 1 jcm-10-05562-f001:**
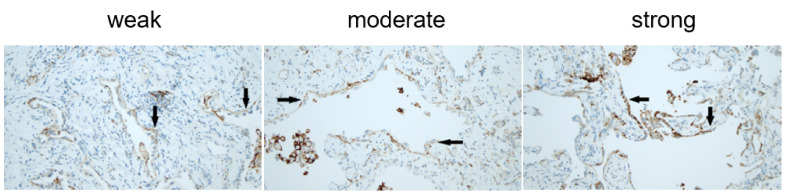
Membranous PD-L1 expression in parenchymal epithelial cells. PD-L1 expression in epithelial cells (arrows). Macrophages are positive and represent an internal control. Medium power.

**Table 1 jcm-10-05562-t001:** Demographic characteristics of patients with IPF and without IPF.

	IPF Patients 43 (44%)	Non-IPF Patients 55 (56%)
Age (median, range)	73 (48–79) *	70 (40–78) *
Gender (male)	26 (60%)	34 (62%)
Smoking %		
- Never	35	36
- Ex	56	51
- Smoker	9	13
Packyears (SD)	30.21 ± 19	25.24 ± 18
FVC % of predicted (SD)	93.60 ± 19	92.73 ± 21
TLco % of predicted (median, IQR)	60 (49–70)	55 (49–66)
6MWT distance (m) (SD)	466 ± 96	491 ± 99
Desaturation ≥ 4 % during 6MWT (SD)	35 (83%) **	35 (71%) **
Traction bronchiectasis	41 (95%) *	35 (64%) *
GAP index		
- 1	34 (79%)	46 (84%)
- 2	9 (21%)	9 (16%)

IPF: idiopathic pulmonary fibrosis; FVC: forced vital capacity; SD: standard deviation; TLco: lung transfer factor for carbon monoxide; IQR: interquartile range; 6MWT: six-minute walk test; m: meters; GAP: gender–age–physiology. * Significantly different (*p* < 0.05). ** Forty-two IPF patients and forty-nine non-IPF patients performed 6MWT.

**Table 2 jcm-10-05562-t002:** Demographic characteristics of patients with PD-L1-positive membrane staining with IPF and without IPF.

	IPF mPD-L1_pos_ *n* = 12	Non-IPF mPD-L1_pos_ *n* = 5
Age (median, range)	76 (67–77)	72 (64–75)
Gender (male)	6 (50%)	3 (60%)
Smoking %		
- Never	58	40
- Ex	33	40
- Smoker	8	20
Packyears (SD)	31 ± 15	37.67 ± 27
FVC % of predicted (SD)	92.31 ± 19	99.83 ± 11
TLco% of predicted (median, IQR)	51 (46–61)	53 (50–61)
6MWT distance (m) (SD)	470 ± 86	506 ± 19
Desaturation ≥ 4 % during 6MWT	9 (75%)	3 (60%)
Traction bronchiectasis	12 (100%) *	2 (40%) *
GAP index		
- 1	7 (58%)	5 (100%)
- 2	5 (42%)	0

IPF: idiopathic pulmonary fibrosis; mPD-L1_pos_: patients positive for membrane-bound programmed death-ligand one; mPD-L1_neg_: patients negative for membrane-bound programmed death-ligand one; FVC: forced vital capacity; SD: standard deviation; TLco: lung transfer factor for carbon monoxide; IQR: interquartile range; 6MWT: six-minute walk test; m: meters; GAP: gender–age–physiology. * Significantly different (*p* < 0.05).

**Table 3 jcm-10-05562-t003:** Demographic characteristics of patients (IPF and non-IPF) with PD-L1-positive and PD-L1-negative membrane staining.

	mPD-L1_pos_ *n* = 17	mPD-L1_neg_ *n* = 81
Age (median, range)	68 (48–79)	68 (40–79)
Gender (male)	9 (53%)	51 (63%)
Smoking %		
- Never	53	32
- Ex	35	57
- Smoker	12	11
Packyears (SD)	33.5 ± 18	26.53 ± 21
FVC, % of predicted (SD)	93.29 ± 17	90.52 ± 20
TLco, % of predicted (median, IQR)	51 (47–61)	60 (49–68)
6MWT distance (m) (SD)	481 ± 74	479 ± 103
Desaturation ≥ 4 % during 6MWT	12 (71%) **	58 (82%) **
Traction bronchiectasis	14 (82%)	62 (77%)
GAP index		
- 1	12 (71%)	68 (84%)
- 2	5 (29%)	13 (16%)

mPD-L1_pos_: patients positive for membrane-bound programmed death-ligand one; mPD-L1_neg_: patients negative for membrane-bound programmed death-ligand one; FVC: forced vital capacity; SD: standard deviation; TLco: lung transfer factor for carbon monoxide; IQR: interquartile range; 6MWT: six-minute walk test; m: meters; GAP: gender–age–physiology. ** All mPD-L1_pos_ patients and 74 mPD-L1_neg_ patients performed 6MWT.

**Table 4 jcm-10-05562-t004:** Demographic characteristics of patients with IPF and PD-L1-positive and PD-L1-negative membrane staining.

	IPF mPD-L1_pos_ *n* = 12	IPF mPD-L1_neg_ *n* = 31
Age (median, range)	76 (67–77)	72 (64–75)
Gender (male)	6 (50%)	20 (64%)
Smoking %		
- Never	58	26
- Ex	33	64
- Smoker	8	10
Packyears	31 ± 15	30.03 ± 21
FVC, % of predicted (SD)	91.33 ± 18	94.03 ± 22
TLco, % of predicted (median, IQR)	51 (46–61) *	63 (55–72) *
6MWT distance (m) (SD)	470 ± 86	464 ± 101
Desaturation ≥ 4 % during 6MWT	9 (75%) **	26 (87%) **
Traction bronchiectasis	12 (100%)	29 (94%)
GAP index		
- 1	7 (58%) *	27 (87%) *
- 2	5 (42%) *	4 (13%) *

IPF: idiopathic pulmonary fibrosis; mPD-L1_pos_: patients positive for membrane-bound programmed death-ligand one; mPD-L1_neg_: patients negative for membrane-bound programmed death-ligand one; FVC: forced vital capacity; SD: standard deviation; TLco: lung transfer factor for carbon monoxide; IQR: interquartile range; 6MWT: six-minute walk test; m: meters; GAP: gender–age–physiology. * Significantly different (*p* < 0.05). ** All mPD-L1_pos_ and 30 of the mPD-L1_neg_ IPF patients performed 6MWT.

**Table 5 jcm-10-05562-t005:** Histological lesions.

	**IPF Patients** ***n* = 43**	**Non- IPF Patients** ***n* = 55**	***p*-Value**
mPD-L1	12 (28%)	5 (9%)	0.015
gPD-L1	33 (76%)	48 (89%)	0.172
Fibroblast foci	17 (40%)	13 (24%)	0.09
Honeycombing	9 (21%)	2 (4%)	0.007
Patchy fibrosis	41 (95%)	43 (78%)	0.016
Patchy fibrosis + fibroblast foci	17 (40%)	12 (22%)	0.057
	**mPD-L1_pos_** ***n* = 17**	**mPD-L1_neg_** ***n* = 81**	***p*-Value**
Fibroblast foci	9 (53%)	21 (26%)	0.028
Honeycombing	2 (12%)	9 (11%)	0.938
Patchy fibrosis	17 (100%)	67 (83%)	0.064
Patchy fibrosis + fibroblast foci	9 (53%)	20 (25%)	0.02
	**IPF mPD-L1_pos_** ***n* = 12**	**Non-IPF mPD-L1_pos_** ***n* = 5**	***p*-Value**
Fibroblast foci	5 (42%)	4 (80%)	0.149
Honeycombing	2 (17%)	0	0.331
Patchy fibrosis	12 (100%)	5 (100%)	
Patchy fibrosis + fibroblast foci	5 (42%)	4 (80%)	0.149
	**IPF mPD-L1_pos_** ***n* = 12**	**IPF mPD-L1_neg_** ***n* = 31**	***p*-Value**
Fibroblast foci	5 (42%)	12 (39%)	0.859
Honeycombing	2 (17%)	7 (23%)	0.669
Patchy fibrosis	12 (100%)	29 (94%)	0.368
Patchy fibrosis + fibroblast foci	5 (42%)	12 (39%)	0.859

IPF: idiopathic pulmonary fibrosis; mPD-L1: membrane-bound programmed death-ligand one; gPD-L1: granular-bound programmed death-ligand one; mPD-L1_pos_: patients positive for membrane-bound programmed death-ligand one; mPD-L1_neg_: patients negative for membrane-bound programmed death-ligand one.

**Table 6 jcm-10-05562-t006:** Changes over one year.

	**IPF** ***n* = 41**	**Non-IPF** ***n* = 50**	***p*-Value**
Change FVC % (SD)	−3.3 ± 10	−1.64 ± 13	0.49
Change TLco % (SD)	−1.5 ± 8	5.2 ± 12	0.003
Change 6MWTD (m) median (IQR)	−12 (−42–13)	−25 (−59–36)	0.14
	**mPD-L1_pos_** ***n* = 15**	**mPD-L1_neg_** ***n* = 76**	***p*-Value**
Change FVC % (SD)	−1.3 ± 11	−2.6 ± 12	0.68
Change TLco % (SD)	0.6 ± 9	2.5 ± 11	0.53
Change 6MWTD (m), median (IQR)	−42 (−97–16)	−15 (−50–29)	0.9
	**IPF mPD-L1_pos_** ***n* = 11**	**IPF mPD-L1_neg_** ***n* = 30**	***p*-Value**
Change FVC % (SD)	−2 ± 8	−3.8 ± 11	0.61
Change TLco %(SD)	−1 ± 7	−1.6 ± 8	0.83
Change 6MWTD (m), median (IQR)	−4 (−97–4)	−13 (−35–19)	0.86
	**IPF mPD-L1_pos_** ***n* = 11**	**Non- IPF mPD-L1_pos_** ***n* = 4**	***p*-Value**
Change FVC % (SD)	−2 ± 8	0.75 ± 18	0.34
Change TLco % (SD)	−1 ± 7	5 ± 13	0.13
Change 6MWTD (m), median (IQR)	−4 (−97–4)	1 (−121–41)	0.73

IPF: idiopathic pulmonary fibrosis; FVC: forced vital capacity; TLco: lung transfer factor for carbon monoxide; 6MWTD: six-minute walk test distance; IQR: interquartile range; mPD-L1_pos_: patients positive for membrane-bound programmed death-ligand one; mPD-L1_neg_: patients negative for membrane-bound programmed death-ligand one.

## Data Availability

All data used during the study are available on reasonable request.

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
