# Peer review of "PD-L1 Expression in Patients with Idiopathic Pulmonary Fibrosis"

_jcm, 2021, doi:10.3390/jcm10235562_

Round 1

Reviewer 1 Report

In the present study, the authors showed the occurrence of mPD-L1 expression in lung tissue in patients with IPF and ILD.

Major comments:

  1. In the text, the authors used DLco however, currently, TLco (lung transfer factor for carbon monoxide) is rather used.
  2. Did the authors adopt the Global Lung Function Initiative (GLI) reference values for PFT measurements? This information is missing in the Methods section.
  3. The authors didn’t mention what statistical software was used for analysis. Additionally, no information is provided about test used for the normality of data distribution.
  4. Line 140, Wilcoxon test, and Mann-Whiney test are two separate tests with different applications. But in the text, it looks as if it is one test.
  5. Usually, values of FVC% pred. and TLco % pred. are rounded to two decimal places. Could the authors explain why decimal places were mostly removed?
  6. Table 1, to expand demographic characteristics the authors could add the information about active smokers, smoking exposure, time since diagnosis, CPI score, and GAP index.
  7. In lines 155-157, the authors provided non-IPF patients' diagnoses however no numbers were given. Whereas in lines 196 and 197 the exact numbers were provided. The reviewer thinks it would be wise to add the missing numbers to demographic characteristics.
  8. The authors indicated changes in table 7, however, no explanation in the Methods section or any other part of this publication, if were they relative changes or absolute changes?
  9. The authors mentioned antifibrotic therapy for IPF patients. The authors provided treatment for non-IPF patients however, IPF is missing. Could the authors provide information pirfenidone or nintedanib therapy was used?
  10. In lines 208-209, the authors mentioned BAL analysis however, no results were presented in the article. If not lymphocytosis, was it connected with the higher number of macrophages in BAL of PD-L1 positive patients?
  11. Lines 221-222, the authors reported a “significant difference in decline in DLCO at one-year follow-up between the patients with IPF (-1.5 ± 8) and the non-IPF patients (5.2 ± 12) (p=0.003)”. The decline is observed in IPF patients, for non-IPF patients, the increase is detected. Based on the presented result the authors observed rather a significant difference in TLco change after one year of treatment between the patients with IPF and the non-IPF patients.
  12. The authors didn’t provide an Institutional Review Board Statement, Informed Consent Statement, Data Availability Statement, Conflicts of Interest.
  13. Could the authors provide the reason why PD-L1 serum level was not measured in IPF and non-IPF patients?

Minor comments:

  1. Spelling check needed.
  2. Not all abbreviation meanings are provided.

Author Response

Major comments:

  1. In the text, the authors used DLco however, currently, TLco (lung transfer factor for carbon monoxide) is rather used.

We thank the reviewer or having made us aware of this and have changed DLco to TLco throughout the paper.

2. Did the authors adopt the Global Lung Function Initiative (GLI) reference values for PFT measurements? This information is missing in the Methods section.

Thank you for asking this question. We have used the GLI reference values for PFT and have added this information to the method section. Please see line 146 -147.

3. The authors didn’t mention what statistical software was used for analysis. Additionally, no information is provided about test used for the normality of data distribution.

Thank you for making us aware of the lack of information. The issues raised above has been added in line 181,182 and 185

4. Line 140, Wilcoxon test, and Mann-Whiney test are two separate tests with different applications. But in the text, it looks as if it is one test.

The Mann-Whitney U test is also called the Wilcoxon-Mann- Whitney test and is not the same as the Wilcoxon signed-rank test. To avoid confusion, we have deleted Wilcoxon in line 183.

5. Usually, values of FVC% pred. and TLco % pred. are rounded to two decimal places. Could the authors explain why decimal places were mostly removed?

In some journals it is standard to add only one decimal, but we have added 2 decimals for all FVC% values. TLco did not have a normal distribution, and median is used with IQR and  therefor there are no decimals .

6. Table 1, to expand demographic characteristics the authors could add the information about active smokers, smoking exposure, time since diagnosis, CPI score, and GAP index.

We have added smoking history and GAP index and explained that patients were seen for diagnostic work-up, thus time since diagnosis is not relevant. We have added time from first visit to cryobiopsy procedure. We do not have data on time from start of symptoms to first visit / diagnosis. CPI is not used at our center and thus we do not have these data.

7. In lines 155-157, the authors provided non-IPF patients' diagnoses however no numbers were given. Whereas in lines 196 and 197 the exact numbers were provided. The reviewer thinks it would be wise to add the missing numbers to demographic characteristics.

We have added numbers of specific non-IPF diagnosis on line 194-196.

8. The authors indicated changes in table 7, however, no explanation in the Methods section or any other part of this publication, if were they relative changes or absolute changes?

Thank you for asking for a clarification of this. We have added that we are using absolute changes to the method section see 6, p 149-150.

9. The authors mentioned antifibrotic therapy for IPF patients. The authors provided treatment for non-IPF patients however, IPF is missing. Could the authors provide information pirfenidone or nintedanib therapy was used?

Thank you for the question. The information on antifibrotic treatment (16 patients had nintedanib, 24 pirfenidone,) has been added to page 14, line 284.

10. In lines 208-209, the authors mentioned BAL analysis however, no results were presented in the article. If not lymphocytosis, was it connected with the higher number of macrophages in BAL of PD-L1 positive patients?

We appreciate the question and have added BAL data in line 260-262.

11. Lines 221-222, the authors reported a “significant difference in decline in DLCO at one-year follow-up between the patients with IPF (-1.5 ± 8) and the non-IPF patients (5.2 ± 12) (p=0.003)”. The decline is observed in IPF patients, for non-IPF patients, the increase is detected. Based on the presented result the authors observed rather a significant difference in TLco change after one year of treatment between the patients with IPF and the non-IPF patients.

Dear reviewer. I am not sure what the question is and have therefore not changed anything.

12. The authors didn’t provide an Institutional Review Board Statement, Informed Consent Statement, Data Availability Statement, Conflicts of Interest.

All approvals required by Danish authorities are mentioned in the methods section on p. 5 line 135- 136 and 7 line 188-192 but has been moved to the end of the paper as per journal style.  Data availability statement, author contribution and conflicts of interest has been added on page 19, line 379-380. COI has also been submitted separately.

13. Could the authors provide the reason why PD-L1 serum level was not measured in IPF and non-IPF patients?

Thank you for your valuable comment. Measuring soluble PD-L1 or lymphocytes expressing PD-L1 in serum as a biomarker is an interesting hypothesis. Some of the earlier studies have found a higher level in IPF patents compared to healthy controls. We have made a biobank with blood samples drawn for each patient at baseline, at one- and two-years follow-up and are planning to make a study investigating this.  

Minor comments:

  1. Spelling check needed.

Spelling check language edition by a professional translator has been performed.

  1. Not all abbreviation meanings are provided.

We have checked and added the missing abbreviations.

Reviewer 2 Report

In this study, Kronborg-White et al. systematically analyse PD-L1 abundance in lung epithelia comparing IPF and non-IPF. This is a timely study and well performed, I suggest only minor modifications:

  1. Please add and discuss PD-L1 in other organs including the kidney (DOI: 10.3389/fimmu.2020.624547). 
  2. It has been shown that PD-L1 positivity is associated with levels of CRP. If available, I would suggest to also correlate these markers in the present patient cohort (DOI: 10.1016/j.suronc.2018.01.002).
  3. With regard of anti-PD-L1/PD1 therapy, adverse effects especially in patients positive for PD-L1 as a potential protective mechanism should also be discussed. 

Author Response

Reviewer number 2

  1. Please add and discuss PD-L1 in other organs including the kidney (DOI: 10.3389/fimmu.2020.624547). 

We have read the article suggested and looked for other interesting articles from other organs without luck. A section has been added in the discussion in line 354-356.

2. It has been shown that PD-L1 positivity is associated with levels of CRP. If available, I would suggest to also correlate these markers in the present patient cohort (DOI: 10.1016/j.suronc.2018.01.002).

We found no association between crp levels and PD-L1 pos patients. This has been added in line 262-263.

3. With regard of anti-PD-L1/PD1 therapy, adverse effects especially in patients positive for PD-L1 as a potential protective mechanism should also be discussed. 

We thank the reviewer for raising this is important point. We have added a section in the discussion on page 18, line 352-356